# Cross-national study of communal attitudes toward individuals with intellectual disabilities in sub-Saharan Africa: Cameroon vs. Ghana

**Maxwell Peprah Opoku**[1]*, **Hala Elhoweris**[1], **Alex Nester Jiya**[2], **Nlem Anne-Paule Ngoh**[2], **William Nketsia**[3], **Emmanuel Opoku Kumi**[4], **Eric Lawer Torgbenu**[5]

**1** Special Education Department, United Arab Emirates University, Al-Ain, United Arab Emirates, **2** Institute of Governance, Humanities and Social Sciences, Pan African University, Yaoundé, Cameroon, **3** School of Education, Western Sydney University, Sydney, Australia, **4** School of Education, University of Tasmania, Launceston, Australia, **5** Department of Physiotherapy and Rehabilitation Sciences, University of Health and Allied Sciences, Ho, Ghana

* Maxwell.p@uaeu.ac.ae

**Data Availability Statement:** The data underlying this study contain sensitive participant information and cannot be shared publicly. Data access

## Abstract

### Background

An intellectual disability (ID) is characterized by a deficit in the functional, cognitive, and adaptive skills required for independent living. Due to the low cognitive capabilities of individuals with IDs, they have become victims of marginalization, exclusion, and denial of their fundamental rights to basic necessities in societies around the world. While efforts are being made to improve service provision to and acceptance of individuals with disabilities, the extent of communal acceptance and recognition of these individuals as equal members of society remains underexplored in sub-Saharan African countries such as Cameroon and Ghana.

### Objective

As attitudes toward individuals with IDs are pivotal in shaping national policies, this cross-national study examined communal attitudes toward persons with IDs in Cameroon and Ghana.

### Method

The Community Living Attitude Scale for Intellectual Disabilities (CLAS-ID) was used to collect data from a total of 741 university students in the two countries. The validity of the scale was assessed using confirmatory factor analysis and principal component analysis. The association between the background variables and attitudes was examined using t-tests, analysis of variance, linear regression, and two-way factor analysis.

### Results

The results showed the validity of the CLAS-ID as a valid tool for measuring communal attitudes toward individuals with IDs in sub-Saharan Africa. The participants appeared

requests can be sent to Emmanuel Barima Agyemang Prempeh at the Ethics Committee of the University of Health and Allied Sciences, Ho- Ghana (eprempeh@uhas.edu.gh).

**Funding:** The author(s) received no specific funding for this work.

**Competing interests:** The authors have declared that no competing interests exist.

ambivalent about attitude towards individuals with ID and other findings showed no association between attitudes and variables such as gender, relation, and contact with individuals with IDs.

## Conclusion

We discuss the need for innovative approaches aimed at changing attitudes toward individuals with IDs in sub-Saharan Africa as well as other study implications.

## Introduction

The capacity of individuals to engage in functional activities, high-order thinking, planning, and decision-making is measured by their intelligence. However, individuals with intellectual disabilities (IDs) are unable to achieve such adaptive functioning due to cognitive deficits or below-average intelligence [1,2], rendering them dependent on others as they are unable to perform basic task requirements for day-to-day living. According to the fifth edition of the *Diagnostic and Statistical Manual of Mental Disorders*, IDs are characterized by deficits in social, conceptual, and adaptive functioning in individuals [1]. The onset of an ID is usually diagnosed before 18 years of age, with leading causes attributed to genetic factors and neurodevelopmental, neurological, and other medical conditions [1,3]. While one to three percent of the population in Western countries live with some form of ID, it is estimated that 10 to 15 individuals in every 1,000 are believed to be living with a form of ID in developing countries [3], including sub-Saharan African countries such as Cameroon and Ghana. From the perspectives of parents and individuals with IDs, empirical evidence suggests that they continue to suffer discrimination and rejection by other people in sub-Saharan African societies [4,5]. For example, parents of children with ID have claimed to be victims of mockery, labelling and some isolated by people in the society [4–8]. Unfortunately, however, studies on community attitudes toward individuals with IDs are rare. With limited formidable policies and facilities available to promote the well-being of persons with IDs in sub-Saharan Africa (e.g., [4,8,9]), there is an urgent need to understand the perceptions of the next frontiers of policymakers, students, toward individuals with IDs in contexts such as Cameroon and Ghana.

Previous studies have mainly reported about the challenges encountered by persons with IDs in society [10–14]. Although some studies have reported favorable dispositions and relationships between persons with IDs and other groups in society [15–18], globally, persons living with IDs are victims of discrimination, marginalization, and exclusion from decision-making on matters concerning their livelihoods [6,13,19–21]. For example, they are denied access to education as teachers hold negative attitudes toward them [22]. In an effort aimed at practicing inclusive education, some teachers are also hesitant to include them because of feelings of inadequacy regarding their teaching skills [14,22–25]. In employment, people with IDs are often overlooked by employers and sidelined from any form of income generation activities [9,11,12]. In terms of sexuality and choosing a partner, people with IDs are also less likely to be considered as spouses by other persons without disabilities [11,26] or may not be considered as having sexual freedom [11,15]. With respect to healthcare, they are often stigmatized and abused by healthcare professionals whose responsibility is to provide accessible health care to them [20,27,28]. In sub-Saharan African countries such as Cameroon and Ghana, local culture interferes with the provision of services to persons with IDs [6,9,19,20,24]. For example, there are strict gender roles which have impact on child rearing and relationship between a

child with disability and family [29,30]. The birth of a child with disability sometimes result in breakdown of homes with female parents likely to be blamed for the onset of the disability as well becoming the primary caregiver of the child [29]. This situation probably necessitates the need to compare attitudes of individuals based on gender to determine whether the situation is improving.

There is intense advocacy aimed at supporting the inclusion of all persons in matters concerning their livelihoods [14,31]. There are also contemporary efforts geared toward the acceptance and inclusion of persons with IDs in public spaces and social services such as education and health care. This advocacy has been led by the United Nations [31], which developed guidelines—in the form of the Convention on the Rights of Persons with Disabilities (CRPD) —aimed at promoting the rights of persons with disabilities, including those living with IDs. For example, Article 1 of the CRPD reiterated the need for countries to respect differences, embrace persons with disabilities, and work toward promoting their inclusion and participation in society. Adherence to the CRPD can be determined by empirical studies on attitudes, which can provide information about whether people in society embrace or are prepared to support persons with IDs to enjoy their fundamental rights as equal members of society.

## Attitudes toward persons with ID

There is considerable research on communal attitudes toward persons with IDs [27,28,32–36]. As a psychological concept, attitude lies at the intersection of beliefs and the outcome of a given behaviour [37–39]. It is an interaction between the beliefs of an individual and their projection about the outcome of a given phenomenon [37]. Once individuals have a favorable belief and conceive the outcome of a behaviour as positive, their attitude toward the behaviour will also be positive [38]. However, attitude toward behaviour is fluid—not static—as evolves over time [23,33]. This has contributed to the development of scales to measure attitudes toward individuals with IDs in society. Some of the standard scales used are the Attitudes towards Intellectual Disability Scale (ATTIP; e.g., [28,33,40,41]) and the Community Living attitude Scale for ID (CLAS-ID; e.g., [27,32,34,36,41–43]).

The CLAS-ID comprises four sub-scales (empowerment, exclusion, sheltering, and similarity) which broader and directly captures some of the issues faced by parents who are raising children with IDs in sub-Saharan Africa. Empowerment refers to the self-belief and capacity of persons with IDs to take ownership and make decisions concerning their lives, while exclusion refers to the extent to which people think that individuals with IDs should be excluded in society [36,44]. Sheltering refers to the help needed by individuals with IDs to keep safe, while similarity has to do with the extent to which people think that persons with IDs are similar to others in society [36,44]. It is important to mention here that the nature of participants could sometimes influence their responses on some of the measure sub-scales such as the sheltering [44]. Where the study tends to focus on psychiatrists, they may be inclined to support the sheltering of persons with ID [44]. However, in the study environment, there are limited facilities to support the development of individuals with ID. This means that the participants may not have their own biases while completing the survey.

The barriers encountered by persons with disabilities are very broad which requires holistic approach to rectify the situation. For example, in sub-Saharan African countries such as Cameroon and Ghana, the exclusion of individuals with IDs with respect to the denial of their fundamental rights, societal rejection, and sub-human categorization [5,7,11,19] closely aligns with the concepts of the CLAS-ID. Also, the scale has yielded appropriate reliability scores in different contexts where they have been used to understand communal attitudes toward individuals with IDs [34–36,41–43]. This lends support for the use of the CLAS-ID in studying

communal attitudes toward individuals with IDs in contexts where it has never been used. As part of efforts toward promoting the inclusion of persons with IDs across sub-Saharan Africa, the study reported here was conducted to provide baseline information and give directions to policymakers about whether attitudes toward persons with IDs are changing.

Students are the future leaders and are expected to play a central role in promoting the acceptance and full inclusion of individuals with IDs in society. Consequently, several studies have sought to understand the attitude of students toward individuals with IDs (e.g., [15,27,41,42,45–48]). Nevertheless, cross-national studies on student attitudes toward persons with IDs are very rare, with few exceptions being Benomir et al. [32] and Sheridan and Scior [42]. Benomir et al. [32] compared students' attitudes toward persons with IDs in Libya and the UK, with the students in Libya recording less favorable attitudes toward persons with IDs than their counterparts in the UK. However, this comparison was conducted between different cultural contexts and countries characterized by other contextual differences. In particular, at the time of the data collection in Libya, there was a civil war in the country, which toppled the then government. Without a central authority and the breakdown of social support systems, the findings reported by the authors were expected. This arguably underscores the need for cross-national studies of countries with a similar culture, experiences, and systems and provides a more justifiable basis for comparison.

In another study, which was conducted by Sheridan and Scior [42] in the UK, the authors compared the attitudes of young White British and British Asians toward individuals with IDs. They reported that British Asians had lower scores on exclusion and higher similarity scores than their White British counterparts. Although different groups with diverse identities were recruited for this study, they shared the same characteristic with respect to living in the same context. In our view, to conduct a cross-national study, the groups being studied should reside in different contexts as this would enable a broader understanding of the findings. With commonalities between countries in sub-Saharan Africa such as Cameroon and Ghana, this study sought to develop a cross-national understanding of student attitudes toward persons with IDs.

There are also inconsistencies between the variables influencing people's attitudes toward persons with IDs [33]. With the exception of contact and relation with a person with an ID, which have been found to be positively associated with attitudes toward persons with IDs (e.g., [18,23,40,42,49–51], inconsistencies have been noted among other background variables. An exception is a study in China that reported no differences between participants in terms of contact and relatives with IDs and their attitudes toward individuals with IDs [36]. With respect to gender, while some studies have reported that males were more positive than females toward persons with IDs (e.g., [33,45]), others have reported otherwise (e.g., [33–35,42,51]). Still, other studies have reported no differences between participants in terms of gender and attitude toward persons with IDs [32,36]. Other studies have reported further inconsistencies regarding general attitudes (positive [33,35,36] or negative [18,27,45] and demographic variables such as age (e.g., [education (e.g., [16], ethnicity [18,42], and religion [34,42]. The fluidity of communal attitudes toward persons with disabilities provides a solid justification for a contextual understanding that could inform national policies and planning. Unfortunately, sub-Saharan African countries' share of research on the communal acceptance and influence of background variables on attitudes remains rare or unresearched.

## The current study

While Cameroon is in central Africa with an estimated population of 26 million [52], Ghana is located in West Africa and has a population of around 30 million [53]. There is a shared

understanding of ID in the two countries. For example, from a traditional perspective, the onset of IDs is linked to the punishment meted out to parents for sins committed, making them unwelcome or "unwanted" in most societies [6,54]. As IDs are linked to abnormality, people with IDs are seldom given any form of assistance by the state. Due to the national conception of ID, they become objects of ridicule, humiliation, and mockery [54,55]. In the context of national development, there is limited consideration of persons with IDs in planning and participation in productive activities [9,54,56–58]. Furthermore, in both countries, teachers in regular classrooms generally reject the idea of practicing inclusive education for students with IDs [9,54,56,58]. They believe that they were not trained to teach students with IDs in regular classrooms [24,25,56], thereby leaving students with IDs to access education in special schools [7,58]. However, there are limited special school facilities to accommodate the growing population of individuals with IDs in both countries [8,58]. It is not surprising that a critical mass of individuals with IDs are not in school and are unlikely to be employed in Cameroon [9,54,58] and Ghana [59,60]. Consequently, parents have to shoulder the burden of raising their children with IDs in hostile social contexts.

Despite the rejection and discrimination perpetuated against persons with IDs in both countries, they are both signatories to the CRPD, which encourages countries to formulate policies or intervention programs aimed at the inclusion of persons with IDs. While Cameroon signed the CRPD in 2008, Ghana ratified it in 2012. In Cameroon, the government passed decree No 2018/6233, which sought to encourage the acceptance of persons with IDs as well as promote their fundamental rights to essential services [61]. In Ghana, the government passed Disability Act 715, 2006, with the sole aim of promoting the fundamental rights and civil liberties of persons with disabilities, such as those living with IDs [62]. However, studies conducted in both countries point to the difficulties faced by persons with IDs and their families in their day-to-day living experiences (e.g., [9,55,57–59]). Research on communal attitudes toward individuals with IDs is very rare in both Cameroon and Ghana, and as students are important to shaping future policies, it is necessary to develop insights into their perspectives toward persons with IDs in both countries. Therefore, the purpose of this study was to develop deeper insight into students' attitudes toward individuals with IDs using the CLAS-ID in the sub-Saharan African context, specifically Cameroon and Ghana, where the tool has never been implemented. The study reported here drew on students from diverse background who somehow mirrors the differences among the people in the society. Indeed, the students are the next frontiers who are expected to be change agents and leaders of public institutions. In view of this, their perception would help to understand whether they would include individuals with ID in national development. Thus, the findings of this study could reflect what patterns in the society with respect to how the society perceive individuals with ID.

We formulated four study hypotheses: first that the CLAS-ID will yield a suitable structure for its use in sub-Saharan Africa; second, that there will be no difference between students on the basis of their country of origin; third, that female students will hold more positive attitudes toward individuals with IDs than their male counterparts; and fourth, that having a relative or contact with an ID will be positively associated with attitudes toward persons with IDs. To confirm or disprove our hypotheses, this cross-national study was guided by the following research questions: 1) What is the underlying factor structure of the CLAS-ID in sub-Saharan African countries such as Cameroon and Ghana? 2) What is the association between students' profile and attitudes toward individuals with IDs in Cameroon and Ghana? 3) What are the predictors of attitudes toward individuals with IDs across Cameroon and Ghana? 4) Does country (as a variable) moderate the relationship between students' profile and attitudes toward persons with IDs?

## Method

### Study participants

The participants of this study were university students pursuing various undergraduate programs in both Cameroon and Ghana. In the view of the authors, this study is the first of its kind to be conducted in sub-Saharan Africa. As such, we recruited students exclusively from public universities where the influence of the state can impact course developments and school policies. Therefore, one public university from each country was selected based on convenience and ease of access of the research team. While the university in Cameroon is mainly social science oriented, the university in Ghana is a medical school for training health professionals. The recruitment process was guided by the following inclusion criteria: The participant a) had to be enrolled in the institution selected for the study; b) had to be at any of the various stages of undergraduate study; c) had a fair knowledge or understanding of IDs; and c) had to be at least 18 years and able to read and sign the consent form.

Printed questionnaires were distributed to the students in both countries. Out of 950 questionnaires distributed, 770 were received by the research team, representing a response rate of 81%. However, 29 questionnaires were excluded because they were partially incomplete or did not indicate the country of data collection. Consequently, 741 questionnaires were retained for this study (see Table 3 for details), of which 413 were recruited from Cameroon and 328 from Ghana. Fifty-one percent of the participants were female compared to 49% male; 71% were between the age of 18 and 21 years compared to 29% who were at least 22 years. While 86% indicated that they had no relative with IDs, 14% indicated that they did.

### Instruments

A two-part questionnaire was used for the data collection. The first part of the instrument collected information about the participants' demographic characteristics, such as country, gender, age, year of study, whether they had a relative with IDs, frequency of contact with people with IDs, and religion. The decision regarding which demographic variables to include in this study was based on a review of the literature (e.g., [18,34,35]).

The second component of the questionnaire was the revised CLAS-ID [63]. The scale comprised 17 items anchored on a five-point scale, with responses ranging from 1 (strongly disagree) to 5 (strongly agree). The scale consisted of four main factors: empowerment, exclusion, sheltering, and similarity. The scale was given to two academics each in Cameroon and Ghana to verify whether the content was appropriate for data collection in the two countries. All the academics responded that the content was appropriate for data collection however, some iterations were suggested which were incorporated in the final draft used for data collection. Some of the items on the scale included the following: "A person would be foolish to marry a person with an intellectual disability," "People with intellectual disabilities should live in special facilities because of the dangers of life in the community," and "People with intellectual disabilities can be productive members of society."

The scale yielded an appropriate reliability score, which was computed using Cronbach's alpha (CLAS-ID = .80). Mean scores—the sum scores divided by the number of items—were reported in this study to facilitate understanding of the results. Thus, a mean score of at least 4 was interpreted as a positive attitude toward persons with IDs.

### Procedure

The study received institutional approvals from the University of Health and Allied Health, Ghana, and the Institute of Governance, Humanities and Social Sciences of the Pan African

University, University of Yaounde II, Cameroon, before the data collection. After the approval, formal letters were sent to the various departments in the schools requesting permission to collect data from the students. While author three collected data in Cameroon, author five collected data in Ghana. Before the lecture, the authors informed the class about the study and waited until the end of the classes to distribute the questionnaires to students who met the inclusion criteria and consented to take part in the study. Before distributing the questionnaire, the study focus, objectives, and the inclusion criteria were explained to the students. They returned the questionnaire one week later when the authors attended the same classes to pick up the questionnaires directly from them.

With respect to the Cameroonian data, the questionnaires were both in English and French since both languages are the medium of instruction and officially recognized languages. Each document given to the students had both English and French versions and the students had an option to either complete the English or the French version. The questionnaire administered in Ghana was in English. The students spent about 20 to 25 minutes to complete each questionnaire. The data were collected between March 2018 and November 2019, and the participants were informed that their participation was voluntary and that there was no financial reward or incentive for completing the questionnaire. They were also told that their identity, department, and class would not be included in the reporting of the study. The questionnaires were distributed to those who met the inclusion criteria and signed the written consent form.

## Data analysis

The questionnaires were first entered into Microsoft Excel for cleaning before being transferred to SPSS, version 26, for analysis. The instrument was then transferred to AMOS, version 27, for computation of the factor analysis. In the first stage, the authors assessed the reliability of the data before proceeding with the data analysis. Using boxplots and histograms, the authors concluded that the data were normally distributed based on a visualization of the outputs.

The authors then proceeded to answer the research questions. To answer research question 1, factor analyses were computed to understand the structural validity of the CLAS-ID. In order to ascertain the validity of the instrument, we followed the format of Su et al. [36] using both confirmatory factor analysis (CFA) and principal component analysis (PCA). According to Worthington and Whittaker [64], the use of CFA and PCA is appropriate to get an accurate understanding of the validity of an adapted scale in a new context. CFA, which was performed using AMOS (version 27), was computed to assess the unidimensionality and validity of the latent constructs of the CLAS-ID. According to Schumacker and Lomax [65], in CFA, the researcher specifies the number of factors and estimates the factor loadings and model parameters before testing the model. For PCA, SPSS was used for the computation. The assumption of inter-reliability correlations was assessed to ensure the suitability of the data for PCA. We then computed mean scores for both the overall attitude scale and sub-scales.

To answer research question 2, independent samples t-tests and analysis of variances (ANOVA) were computed for the demographic variables, with two and at least three levels, respectively. The t-tests and ANOVAs were computed to understand the association between the demographic variables and attitudes toward persons with IDs. Here, we used Levene's test to ensure that the assumption of homogeneity of variance was not violated. For the t-tests, in the event of violation, the results of the equal variances assumed, which were supposed to be reported, were not considered. However, the equal variances not assumed were reported. For the ANOVAs, in the event of violation, the results of the Welch statistics were reported. To determine the weight of both the t-tests and ANOVAs, we computed partial eta-squared. The

effect sizes were interpreted as follows: small = .01 to .05; moderate = .06 to .9; and large = at least 1 [66].

To answer research question 3, a linear regression was computed to ascertain the predictors of attitudes. Before computing the linear regression, we used Pearson's moment co-efficient correlations to understand the associations between the sub-scales. Although our intention was not to determine causative effect, we sought to understand the strength of the associations. The correlations were interpreted as follows: small (r = .10 to .29); medium (r = .30 to .49); and large (r = .50 to 1.0) [66]. Afterwards, we proceeded to compute the predictors of attitudes using linear regressions. An initial analysis showed that there was no violation of the following assumptions: assumptions of normality, linearity, multicollinearity, and homoscedasticity.

To answer research question 4, a two-way factor analysis was used to determine the moderation effect of country on attitude toward persons with IDs. We checked the output for Levene's test to ensure that the homogeneity of the variances was not violated. Again, we assessed the effect size using partial eta-squared.

## Results

Overall, the students had ambivalent attitudes (*M* = 3.25; *SD* = 0.58) toward persons with IDs. With the exception of similarity, which showed that the students were slightly positive toward persons with IDs (*M* = 4.66; *SD* = 1.20), the rest showed either ambivalence (empowerment, *M* = 3.28; *SD* = 1.10; sheltering, *M* = 3.95; *SD* = 1.08) or negative attitudes (exclusion; *M* = 2.24; *SD* = .91).

### Structural validity of CLAS-ID

First, CFA was computed for 17 observed variables used to assess attitudes toward persons with IDs in Cameron and Ghana. As shown in Fig 1 and Table 1, the results point to four latent factors in reflecting the associated observed variables. These factors were exclusion, similarity, empowerment, and sheltering.

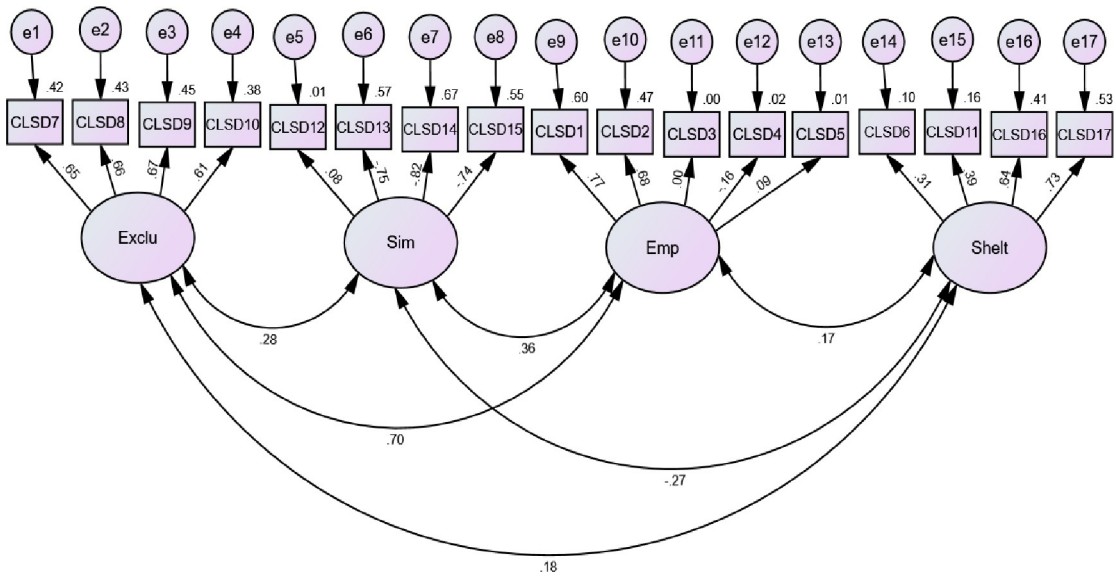

**Fig 1. CFA of CLAS-ID scale.**

**Table 1. Standardised regression weights of CLSD scale.**

| Items | Factors | Estimate |
| --- | --- | --- |
| CLSD7 | <--- Exclusion | .647 |
| CLSD8 | <--- Exclusion | .657 |
| CLSD9 | <--- Exclusion | .668 |
| CLSD10 | <--- Exclusion | .615 |
| CLSD1 | <--- Empowerment | .773 |
| CLSD2 | <--- Empowerment | .684 |
| CLSD3 | <--- Empowerment | .004 |
| CLSD4 | <--- Empowerment | -.156 |
| CLSD5 | <--- Empowerment | .086 |
| CLSD12 | <--- Similarity | .082 |
| CLSD13 | <--- Similarity | -.753 |
| CLSD14 | <--- Similarity | -.821 |
| CLSD15 | <--- Similarity | -.743 |
| CLSD6 | <--- Sheltering | .313 |
| CLSD11 | <--- Sheltering | .394 |
| CLSD16 | <--- Sheltering | .644 |
| CLSD17 | <--- Sheltering | .730 |

The standardized regression weights presented in Table 1 ranged from -.821 to .773, suggesting that some of them were far below the acceptable threshold of 0.5 (see [67]). In addition, the model produced a squared multiple correction ranging from 0.000 to 0.675. As reported by Awang [67], in CFA, all items should have a squared multiple correlation of 0.4 or higher. This suggests that some of the items in the present model were below standard.

Several goodness-of-fit indicators can be used to test the adequacy of a CFA model (e.g., [62,67–69]). For example, Awang [67] reported that goodness of fit is evaluated by employing the chi-square ($\chi$2) (P > .05), the comparative fit index (CFI > .90), the Tucker–Lewis Index (TLI > .90), and the root mean-square error of approximation (RMSEA < .08).

The chi-square value was significant ($\chi^2$ (113) = 837.36, $p$ = .001). The other model fit indices were as follows: CFI = 0.80; TLI = .70; and RMSEA = .080. These results suggest that the model did not fit the data acceptably in the population since all the indicators fell outside the commonly agreed threshold.

Second, the 17-item CLAS-ID was subjected to PCA. The assumptions of inter-reliability correlations were assessed to ensure its suitability for PCA. The output inspection showed that most had a correlation co-efficient of .30. Also, the Kaiser-Meyer-Oklin value was .77, suggesting its appropriateness. Further, the Bartlett test of sphericity was statistically significant at .001.

The PCA showed that four components had eigenvalues exceeding 1. An inspection of the scree plot showed four clear breaks. Since at least three items loaded on each factor, we decided to retain the four factors of the CLAS-1D. The four-component factors explained a total of 54% of the variance, with factors I, II, III, and IV contributing 21%, 16%, 10%, and 7%, respectively. The communality values for all the items were above .03, which means that all the items loaded properly on the factors.

To further support our analysis, an oblimin rotation was conducted to compare or support our interpretation. The rotation supported the four above-identified factors. For the oblimin rotation, the Kaiser-Meyer-Oklin was .77, and Bartlett's test of sphericity was significant at .001. The four factors loaded strongly, with all exceeding the .3 threshold. The communality values also exceeded .3 for all the items.

We determined the names of the factors based on their comparability with the original scale (See Table 2). Thus, Factor I was called exclusion (7 items = 1, 2, 7, 8, 9, 10, 12), Factor II similarity (3 items = 13, 14, 15), Factor III empowerment (3 items = 3, 4, 5), and Factor IV sheltering (4 items = 6, 11, 16, 17).

## Student profile and attitudes

Independent samples t-tests were conducted to compare the association between the demographic profile of students and their attitudes toward persons with IDs (see Table 3). On overall attitudes, significant differences were found between attitudes and two demographics variables, country and age. However, students in Cameroon were more receptive of persons with IDs than their counterparts in Ghana, $t$ (739) = 3.02, $p$ = .002, with a small effect size (partial eta-squared = .01). With respect to age, younger students in both countries reported more positive attitudes toward persons with IDs than older students, $t$ (739) = 3.00, $p$ = .003, with a very small effect size (partial eta-squared = .01).

Other interesting differences were identified in the sub-scales. With respect to gender and having a relative with an ID, there were differences between the participants in terms of exclusion and similarity. Regarding exclusion, male students had more positive attitudes than female students. However, the female students were more positive than their male counterparts in terms of similarity.

With respect to exclusion, the participants who indicated that they had a relative with an ID were more positive than those who indicated otherwise. However, regarding similarity, those

**Table 2. Summary of factor loading of CLAS-ID.**

|  | Item | I | II | III | IV |
|---|---|---|---|---|---|
| 1 | People with intellectual disability should not be allowed to marry and have children | .62 |  |  |  |
| 2 | A person would be foolish to marry a person with intellectual disability | .59 |  |  |  |
| 7 | Increased spending on programs for people with intellectual disability is a waste of money | .74 |  |  |  |
| 8 | Homes and services for people with intellectual disability downgrade the neighbourhoods they are in | .72 |  |  |  |
| 9 | People with intellectual disability are a burden on society | .73 |  |  |  |
| 10 | Homes and services for people with intellectual disability should be kept out of residential neighbourhoods | .70 |  |  |  |
| 12 | People with intellectual disability do not need to make choices about the things they will do each day | .45 |  |  |  |
| 13 | People with intellectual disability have goals for their lives like other people |  | .84 |  |  |
| 14 | People with intellectual disability can be productive members of society |  | .80 |  |  |
| 15 | People with intellectual disability can have close personal relationships just like everyone else |  | .81 |  |  |
| 3 | People with intellectual disability can plan meetings and conferences without assistance from others |  |  | .78 |  |
| 4 | People with intellectual disability can be trusted to handle money responsibly |  |  | .69 |  |
| 5 | The opinion of a person with intellectual disability should carry more weight than those of family members and professionals in decisions affecting that person |  |  | .74 |  |
| 6 | Special or separate workshops for people with intellectual disability are essential |  |  |  | .57 |
| 11 | People with intellectual disability need someone to plan their activities for them |  |  |  | .51 |
| 16 | People with intellectual disability should live in special facilities because of the dangers of life in the community |  |  |  | .72 |
| 17 | People with intellectual disability usually should be in group homes or other facilities where they can have the help and support of staff |  |  |  | .76 |

I = Exclusion; II = Similarity; III = Empowerment; IV = Sheltering.

**Table 3. Association between demographics and attitudes.**

| Category (N = 741) | Sample (%) | Attitude Total | Exclusion | Similarity | Empowerment | Sheltering |
|---|---|---|---|---|---|---|
| *Country* | | | | | | |
| Cameroon | 413 (56%) | 3.31 (.50) | 2.21 (.80) | 4.84 (1.01) | 3.16 (1.04) | 4.20 (1.06) |
| Ghana | 328 (44%) | 3.18 (.67) | 2.27 (1.03) | 4.44 (1.38) | 3.43 (1.15) | 3.63 (1.01) |
| *t* | | 3.02**# | -.86 | 4.56** | -3.33**# | 7.43** |
| partial eta squared | | .01 | .001 | .03 | .02 | .07 |
| *Gender* | | | | | | |
| Male | 364 (49%) | 3.25 (.58) | 2.32 (.91) | 4.55 (1.23) | 3.27 (1.13) | 3.89 (1.07) |
| Female | 377 (51%) | 3.26 (.58) | (.90) | 4.78 (1.17) | 3.30 (1.07) | 4.01 (1.09) |
| *t* | | -.20 | 2.47** | -2.63** | -.38 | -1.60 |
| partial eta squared | | .001 | .008 | .009 | .001 | .003 |
| *Relative with ID* (n = 740) | | | | | | |
| Yes | 107 (14%) | 3.26 (.52) | 2.40 (0.79) | 4.36 (1.34) | 3.22 (1.04) | 3.95 (1.04) |
| No | 633 (86%) | 3.25 (.59) | 2.20 (.91) | 4.71 (1.17) | 3.28 (1.11) | 3.95 (1.09) |
| *t* | | .14 | 2.15* | -2.59*# | -.52 | -.04 |
| partial eta squared | | .01 | .008 | .01 | .001 | .001 |
| *Age* | | | | | | |
| 17–21 | 528 (71%) | 3.29 (.58) | 2.19 (.90) | 4.80 (1.12) | 3.33 (1.12) | 4.07 (1.10) |
| 22 years and more | 213 (29%) | 3.15 (.58) | 2.36 (.91) | 4.32 (1.34) | 3.15 (1.04) | 3.66 (0.98) |
| *t* | | 3.00** | -2.35* | 4.65**# | 2.03* | 4.67** |
| Partial eta squared | | .01 | .007 | .03 | .006 | .03 |
| *Year of study* (n = 740) | | | | | | |
| Year 1 | 136 (18%) | 3.21 (.60) | 2.11 (.85)ᵃ | 4.73 (1.31)ᵃ | 3.31 (1.22) | 3.93 (1.07)ᵃ |
| Year 2 | 139 (19%) | 3.35 (.62) | 2.34 (.98)ᵃ,ᵇ | 4.74 (1.06)ᵃ | 3.40 (1.15) | 4.03 (1.07)ᵃ |
| Year 3 | 306 (41%) | 3.30 (.51) | 2.36 (.91)ᵇ | 4.42 (1.23)ᵇ | 3.19 (1.05) | 3.78 (1.10)ᵃ,ᵇ |
| Year 4 | 159 (22%) | 3.25 (.58) | 2.02 (.84)ᶜ | 5.02 (1.07)ᵃ,ᶜ | 3.33 (1.05) | 4.23 (1.02)ᵃ,ᶜ |
| *F* | | 2.49 | 6.69** | 9.48** | 1.27## | 6.29** |
| Partial eta squared | | .01 | .03 | .04 | .005 | .03 |
| *Contact with ID* | | | | | | |
| Frequently | 208 (28%) | 3.31 (.57) | 2.31 (.96)ᵃ | 4.69 (1.17) | 3.32 (1.15) | 4.02 (1.07) |
| Occasionally | 172 (23%) | 3.19 (.54) | 2.09 (.81)ᵇ | 4.69 (1.17) | 3.25 (.99) | 3.96 (1.08) |
| Rarely | 361 (49%) | 3.25 (.60) | 2.27 (.91)ᵃ,ᵇ | 4.64 (1.24) | 3.28 (1.12) | 3.90 (1.09) |
| *F* | | 2.00 | 3.18* | .20 | .23# | .83# |
| Partial eta squared | | .005 | .009 | .001 | .001 | .002 |
| *Religion* | | | | | | |
| Christian | 492 (66%) | 3.26 (.57)ᵃ | 2.24 (.90) | 4.68 (1.18)ᵃ | 3.30 (1.09)ᵃ | 3.96 (1.07)ᵃ |
| Muslim | 71 (10%) | 3.02 (.60)ᵇ | 2.24 (.80) | 4.29 (1.39)ᵇ | 2.85 (1.03)ᵇ | 3.55 (1.11)ᵇ |
| African Tradition | 178 (24%) | 3.32 (.58)ᶜ | 2.22 (.97) | 4.76 (1.18)ᵃ,ᶜ | 3.40 (1.14)ᵃ,ᶜ | 4.08 (1.06)ᵃ,ᶜ |
| *F* | | 7.09** | .03 | 4.18* | 6.59** | 6.32** |
| Partial eta squared | | .02 | .001 | .01 | .02 | .001 |

*P < .05

**P < .01

ID = Intellectual Disability; superscripts (a,b,c) shows differences.

who said that they did not have a relative with an ID were more positive than those who indicated otherwise. Further, one-way between-groups ANOVAs were conducted to explore the association between three levels of demographics and attitudes toward persons with IDs

**Table 4. Summary of demographics regressed on attitude.**

| Predictor | B | SE | Beta | t | P |
|---|---|---|---|---|---|
| Country | -1.61 | .78 | -.08 | -2.10 | .04* |
| Gender | -.19 | .76 | -.01 | -.25 | .81 |
| Age | -2.01 | .89 | -.09 | -2.26 | .02* |
| Religion | .11 | .43 | .01 | .25 | .80 |
| Year of Study | .35 | .38 | .04 | .92 | .36 |
| Relative with Disability | .19 | .99 | .01 | .19 | .85 |
| Contact with Person with ID | -.37 | .44 | -.03 | -.85 | .40 |

*P < .05

**P < .01; ID = Intellectual Disability.

(see Table 3). On overall attitudes, there were significant differences between the participants only in terms of religion, $F$ (2, 738) = 7.09, $p$ = .001. However, the effect size was small (partial eta-squared = .02). A post-hoc comparison using the Tukey HSD test indicated that students who were African traditionalists were more positive than those who were Christians and Muslims.

## Predictors of attitudes

The associations between the sub-scales were computed using Pearson's correlation co-efficient. While there was a correlation between exclusion and similarity ($r$ = -232, $p$ = .001) and exclusion and sheltering ($r$ = .19, $p$ = .001), there was no correlation between exclusion and empowerment ($r$ = .04, $p$ = .35). Also, there was a positive but small correlation between similarity and empowerment ($r$ = .25, $p$ = .001) and similarity and sheltering ($r$ = .23, $p$ = .001). In addition, while there was no correlation between empowerment and sheltering ($r$ = .03, $p$ = .42), there was a very small correlation between sheltering and exclusion ($r$ = .19, $p$ = .001).

The predictors of attitudes toward individuals with IDs were assessed using linear regression (see Table 4). The demographic variables contributed only two percent to the variance in attitudes, $F$ (7, 739) = 2.25, $p$ = .03. In terms of the demographics, only gender ($beta$ = -.25, $p$ = .04) and age ($beta$ = -2.26, $p$ = .02) contributed significantly to the variance in attitudes.

## Effects of country and demographics on attitudes

A two-way between-groups ANOVA was computed to assess the impact of country and other demographics on attitudes, as measured by the CLAS-ID (see Table 5). Country interacted only with religion. The interaction effect between country and religion was statistically significant, $F$ (2, 735) = 4.64, $p$ = .01, with a very small effect size (partial eta-squared = .004). Overall,

**Table 5. Summary of two-way analysis of variance.**

| Variable | MS | F | p | Partial eta squared |
|---|---|---|---|---|
| Gender | 82.08 | .85 | .36 | .01 |
| Age | 324.14 | 3.38 | .07 | .01 |
| Year of Study | 229.79 | 2.40 | .07 | .01 |
| Religion | 436.95 | 4.64 | .01** | .004 |
| Relative with ID | 432.12 | .65 | .42 | .01 |
| Contact with Persons with ID | 94.90 | .98 | .38 | .01 |

**P < .01; ID = Intellectual Disability.

the African traditionalists ($M = 3.32$; $SD = .58$) were more positive than those in other religions (Christians, $M = 3.26$, SD = .57; Muslims, $M = 3.02$, $SD = .60$). However, a post-hoc comparison using Tukey HSD showed a significant difference between Muslims and the other two religions, which did not differ from each other. In both countries, the African traditionalists appeared slightly more receptive toward individuals with IDs than those in the other religions.

## Discussion

In this study, we attempted to explore communal attitudes toward persons with IDs in two sub-Saharan African countries, Cameroon and Ghana. The CLAS-ID, which has been used in various contexts to measure communal attitudes toward individuals with IDs, was adopted for this study. To the best of our knowledge, this is the first time that the CLAS-ID has been used to study communal attitudes toward persons with ID in the sub-Saharan African context. Thus, both CFA and PCA were computed to understand the structural validity of the scale. Although CFA did not support the validity of the scale in studying attitudes toward persons with IDs, PCA supported and confirmed the four-factor structure of the scale. This finding is inconsistent with a previous result of a two-structure model in a study of communal attitudes toward individuals with IDs in a Chinese context [36]. The structural validity of the CLAS-ID confirmed our earlier hypothesis that the tool would be a valid instrument to study communal attitudes toward individuals with IDs in sub-Saharan African contexts. The challenges faced by individuals with IDs are enormous, and the items on the CLAS-ID reflect some of the issues confronting them or serving as barriers to their inclusion in the two societies. For example, there are discussions concerning their discrimination [6–8,11–13,28], their inability to make or participate in decision makings [58], a strong preference for their education in segregated environments [7,8], as well as reluctance from the community to accept them as equal members of society [7,11]. These challenges faced by persons with IDs and their families are arguably closely aligned to the items on the CLAS-ID. This study has provided support for the use of the CLAS-ID to study communal attitudes toward individuals with IDs in African contexts.

Positive attitudes toward persons with IDs are an indication that the community is willing to support the inclusion of these individuals in the society [33]. This has prompted the need for empirical studies to understand the nature of communal attitudes so as to develop plans to help individuals with IDs enjoy their fundamental rights in society. In the study reported here, the study participants were ambivalent about their attitudes toward individuals with IDs. This finding was inconsistent with the results of previous studies that have reported positive [33,35,36] or negative [18,27,45] attitudes toward individuals with IDs. The results of this study may not be surprising due to the persistent rejection of and discrimination toward persons with IDs in both Cameroon and Ghana [7,9]. The most concerning aspect is that university students—who are expected to take the lead and reform existing policies and practices toward attaining a better life for individuals with IDs—were unsure about their attitudes. Although the participants seemed to indicate that individuals with IDs were equal to them, and they did not support their exclusion, they were unsure about their ability to make decisions to ensure their safety. This arguably suggests that individuals with IDs may still be at risk of rights abuses and denial of access to basic services that would enable them to be self-confident and take ownership of matters concerning their lives. These results call for more education and the development of courses on IDs that would best enable the participants to promote the well-being of individuals with IDs in the two societies.

The findings on the association between country and attitudes toward individuals with IDs did not support our hypothesis. Although we hypothesized that there would be no difference between the participants based on country of origin, the results showed otherwise. Indeed, the

participants from Cameroon indicated more favorable attitudes toward persons with IDs than their counterparts in Ghana. While the participants from the two countries had a similar understanding of ID and commonality in terms of the barriers that people with IDs encounter, the Cameroonian students appeared to have more supportive attitudes than their Ghanaian counterparts. This finding is partly consistent with those of other cross-national studies on differences between participants based on their origin or ethnicity [18,32,34,42]. In our view, such differences could be attributed to the background or the program of study of the study participants. While the Ghanaian participants were medical students enrolled in programs such as medicine, nursing, and other allied health professions, the Cameroonian students were mainly in the social sciences and were pursuing degree programs such as sociology and anthropology. Arguably, while students in the medical field might think of the etiology of IDs, those from the social sciences might be thinking of how society could support persons with IDs in overcoming systemic barriers. Although our explanation could shed light on the differences between the participants, future qualitative studies could explore students' understanding of IDs to ascertain whether their study programs could influence their attitudes.

Gender has consistently emerged as an important variable in influencing attitudes toward individuals with IDs [18,33,34]. In this study, our hypothesis that female students would have more positive attitudes toward individuals with ID was not supported by the results. The study finding showed no significant difference between the participants with respect to gender. While this finding is consistent with that of previous studies [32,36], it is inconsistent with other studies that have reported significant differences between male and female participants in terms of their attitudes toward individuals with IDs [33–35,51]. However, there were notable differences between the participants on exclusion and similarity. While the male students seemed more inclined to support the exclusion of individuals with IDs, the female students indicated that individuals with IDs were more similar to other persons in society. This finding is unsurprising because, in the African context, females seem to be more sympathetic toward persons with IDs than males [70]. This has been attributed to the gender roles in society [70]. In African countries, females oversee child-raising and, as such, have closer relationships with, and are sympathetic toward the plight of, children. Studies have reported that, in most instances, when a child with a form of disability is born in Africa, males normally abandon their children or refuse to take part in raising them [29,30]. The motherly role seemed to have influenced the perception of the student participants in terms of being low on exclusion and claiming individuals with IDs as equals.

An interesting observation was made on the association between attitude and having a relative with an ID. Our hypothesis was not supported by the study findings as there were no significant differences between the participants in terms of attitudes and having a relative with an ID. This finding is inconsistent with that of previous studies that have mainly reported a significant relationship between having a relative with an ID and attitude toward individuals with IDs [18,28,33,42,51]. Although our hypothesis was not supported, a curious observation was made on the differences between the participants on two sub-scales. In particular, participants without a relative with an ID were less supportive of exclusion than those who had a relative with an ID. Also, participants without a relative with an ID self-reported that they were more similar to persons with IDs than those who had relatives with IDs. This finding is peculiar because, in most cases, those who have relations with a person with an ID would be supportive of their inclusion and recognition as equal members of society. However, in this study, participants with a relative with an ID seemed to support the exclusion of this relative, possibly even failing to recognize them as equal members of society. In our view, this could be linked to the hostile attitude and the stigma attached to having a family member with an ID. The family also suffers marginalization, mockery, and abuse from the wider society because of their

association with a member with an ID [5–7,9,20,55]. This sometimes compels family members to hide their member with disabilities from public view [29]. This could explain why those participants with a relative with an ID were more supportive of their exclusion. This finding suggests the need for policymakers to expedite public education about IDs so as to enhance the acceptance of individuals with IDs in the wider society and address their rejection.

Another finding that did not support our hypothesis was that pertaining to frequent contact. We hypothesized that frequent contact with individuals with IDs could be positively associated with attitudes. However, our finding demonstrated no significant relationship between contact and attitude toward persons with IDs. This finding is inconsistent with that of previous studies that have reported a consistently positive association between attitudes and contact with individuals with IDs [27,33,42]. Indeed, there was no difference in attitudes between those who indicated that they had frequent contact with persons with IDs and those indicating otherwise. It has been argued that regular contact with individuals with IDs enables others to understand the uniqueness of such individuals and change their perceptions of them [15]. Unfortunately, this did not seem to be the case in Cameroon and Ghana, where more regular contact had no impact on individual attitudes. There is a possibility that the negative societal perceptions toward persons with ID are so entrenched that a deliberate and persistent approach is required to achieve parity in society. This arguably explains the enormity of the task ahead of policymakers in charge of creating a conducive environment for individuals with IDs. Nevertheless, a more innovative approach to changing public perceptions of persons with IDs, such as having a course on disability at all levels of education, might be required in Cameroon and Ghana.

## Study limitations

A number of study limitations could affect the interpretation of the findings reported here. First, the data comprised self-evaluations from participants who responded to a standard international tool used for measuring communal attitudes toward individuals with IDs. It was beyond the scope of this study to ascertain or observe the actual reaction of the participants in their encounters with persons with IDs. However, adequate information was provided to the participants before they took part in this study, and we are confident that they provided appropriate responses to the items on the scale. Second, the CFA did not confirm the structural validity of the CLAS-ID in this study. Although it supported the four-factor structure, the measurement indices did not reach the required thresholds. Previous studies have encountered similar situations where they used only PCA to validate the factor structure of the CLAS-ID [29]. Nonetheless, our study confirmed the four-factor structure of the CLAS-ID, and we recommend its use as a valid tool to measure communal attitudes toward individuals with IDs. Third, one institution was selected in each country. Therefore, the results may not be representative of the views of all university students in both countries. With the internationalization of education and diversity among students, we contend that the profile of the students may not be different from what may obtain in other institutions. Indeed, this study has filled a major gap in the literature through the use and validation of the CLAS-ID in the sub-Saharan African context (Cameroon and Ghana).

## Conclusion and implications of the study

In this study, we used the CLAS-ID to assess the attitudes of students toward individuals with IDs in two sub-Saharan African countries. While a large body of literature has reported on the fluidity of attitudes toward individuals with IDs in mainly Western and Asian countries, a study of this kind was novel in the sub-Saharan Africa context. Generally, the study

participants were ambivalent in their attitudes toward persons with IDs across both countries. Additionally, the findings of the study only confirmed one of the four study hypotheses. To begin with, the study confirmed the use of the CLAS-ID as a valid tool for measuring attitudes toward individuals with IDs in African contexts. The PCA confirmed the four-factor structure of the revised CLAS-ID developed by Henry et al. [63]. However, our hypotheses on country, gender, and having a relative or friend with an ID were not supported by the study findings. Also, our findings differed from those of previous studies on attitudes toward persons with disabilities.

Especially in terms of gender, having a relative or contact with a person with an ID had no association with attitude. However, other interesting trends emerged in the study. For example, while the male students favored more exclusion of persons with IDs, the female students indicated that there were more similarities between individuals with IDs and other persons in society. Also, participants with relatives with IDs or those who had frequent contact with a person with an ID were more in favor of their societal exclusion. The findings of this study have added to the inconsistencies about the association between background variables and communal attitudes toward persons with IDs.

The participants of this study were students from heterogenous backgrounds who probably mirror the diversity in the society. This probably lends support for consideration of the findings of this study in effort toward enhancing the well-being of individuals with ID in the society. The findings point to the need for more work and advocacy to promote the rights and well-being of persons with IDs in Cameroon and Ghana. Although students are future leaders who could influence policies and reform social norms in favor of more support for persons with IDs, the student participants seemed uncertain about their attitudes. Importantly, in both countries, society harbors negativity toward persons with disabilities. This is evident in their lack of acceptance and consideration of the needs of persons with IDs. However, the findings of this study offer less hope in terms of the readiness of future leaders in changing the status of persons with IDs and advocating more support for them. Policymakers could consider more innovative approaches to public education and sensitization to the capabilities and rights of persons with IDs. With education serving as the bedrock of social change, policymakers could consider introducing courses in disability at all levels of education. This will enable future leaders, such as those who took part in this study, to change their attitudes and be aware of the rights of individuals with IDs. Without such an approach, there is uncertainty as to whether society may be able to accord people with IDs respect and recognition as equal members.

## Acknowledgments

### Declarations

The authors wish to thank all the students who took part in this study.

## Author Contributions

**Conceptualization:** Maxwell Peprah Opoku, Hala Elhoweris, Alex Nester Jiya, Nlem Anne-Paule Ngoh, William Nketsia, Emmanuel Opoku Kumi, Eric Lawer Torgbenu.

**Data curation:** Maxwell Peprah Opoku, Alex Nester Jiya, Nlem Anne-Paule Ngoh, William Nketsia, Emmanuel Opoku Kumi, Eric Lawer Torgbenu.

**Formal analysis:** Maxwell Peprah Opoku, Hala Elhoweris, William Nketsia, Emmanuel Opoku Kumi, Eric Lawer Torgbenu.

**Investigation:** Maxwell Peprah Opoku, Alex Nester Jiya, Nlem Anne-Paule Ngoh, William Nketsia, Emmanuel Opoku Kumi.

**Methodology:** Maxwell Peprah Opoku, Hala Elhoweris, Alex Nester Jiya, Nlem Anne-Paule Ngoh, William Nketsia, Emmanuel Opoku Kumi, Eric Lawer Torgbenu.

**Project administration:** Alex Nester Jiya, Nlem Anne-Paule Ngoh, William Nketsia, Eric Lawer Torgbenu.

**Writing – original draft:** Maxwell Peprah Opoku, Hala Elhoweris, Alex Nester Jiya, Nlem Anne-Paule Ngoh, William Nketsia, Emmanuel Opoku Kumi, Eric Lawer Torgbenu.

**Writing – review & editing:** Maxwell Peprah Opoku, Hala Elhoweris, Alex Nester Jiya, Nlem Anne-Paule Ngoh, William Nketsia, Emmanuel Opoku Kumi, Eric Lawer Torgbenu.

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
