## [Decision Letter · Decision Letter 0]

9 Jun 2021

PONE-D-21-15410

Cross-national Study of Communal Attitudes toward Individuals with Intellectual Disabilities in sub-Saharan Africa: Cameroon vs. Ghana

PLOS ONE

Dear Dr. Opoku,

Thank you for submitting your manuscript to PLOS ONE. I have read your paper and I have received reviews from 2 experts. They agree that this paper is interesting, but also believe the paper may be improved through a revision. I concur with their views. The comments are straightforward and you should have no problems in addressing them. Therefore, we invite you to submit a revised version of the manuscript that addresses the points raised during the review process.

We look forward to receiving your revised manuscript.

Kind regards,

Robert Didden

Academic Editor

PLOS ONE

Journal Requirements:

3. PLOS requires an ORCID iD for the corresponding author in Editorial Manager on papers submitted after December 6th, 2016. Please ensure that you have an ORCID iD and that it is validated in Editorial Manager. To do this, go to ‘Update my Information’ (in the upper left-hand corner of the main menu), and click on the Fetch/Validate link next to the ORCID field. This will take you to the ORCID site and allow you to create a new iD or authenticate a pre-existing iD in Editorial Manager. Please see the following video for instructions on linking an ORCID iD to your Editorial Manager account: https://www.youtube.com/watch?v=_xcclfuvtxQ.

4. Please ensure that you include a title page within your main document. You should list all authors and all affiliations as per our author instructions and clearly indicate the corresponding author.

Additional Editor Comments:

Reviewers' comments:

Reviewer's Responses to Questions

**Comments to the Author**

1. Is the manuscript technically sound, and do the data support the conclusions?

Reviewer #1: Yes

Reviewer #2: Yes

2. Has the statistical analysis been performed appropriately and rigorously? 

Reviewer #1: Yes

Reviewer #2: I Don't Know

3. Have the authors made all data underlying the findings in their manuscript fully available?

Reviewer #1: No

Reviewer #2: No

4. Is the manuscript presented in an intelligible fashion and written in standard English?

Reviewer #1: Yes

Reviewer #2: Yes

5. Review Comments to the Author

Reviewer #1: Very well done article: I have the following suggestions:

1. There has been some research about the CLAS-ID being mainly useful for university students, who it was developed for, and not lay people or health professionals. Maybe comment on the applicability of the study results to the general population and/or government stakeholders.

2. One limitation of the CLAS-ID is that there are conflicting findings about the purpose and value (positive or negative) of the sheltering subscale. Please comment: Ouellette-Kuntz et al. 2003 and Ouellette-Kuntz et al. 2012 comment on this, I believe.

3. In the introduction, provide stronger rationale for looking at differences by gender, as that was a key study hypothesis.

4. In the introduction, provide stronger rationale for the Theory of Planned Behaviour, and comment on its other constructs (subjective norms, perceived behavioural control) or omit.

5. What program or discipline were the students in? Different groups of students (e.g. law vs. social care students) have been shown to have different attitudes.

6. Comment on whether gathering data from students at the end of lecture could have influenced participation, and whether participation in the study was voluntary or for course credit.

Reviewer #2: I would like to thank the authors for exploring an interesting and important topic. The study explores the attitudes towards people with intellectual disability in the geographical context of Cameroon and Ghana while investigating psychometric properties of a questionnaire--the community living attitude scale for intellectual disability (CLAS-ID)--measuring the attitudes towards people with intellectual disability (CLAS-ID). The study recruits 741 university students from two public universities in Cameroon and Ghana, selected based on convenience. The study concluded the CLAS-ID is a sound instrument to measure attitudes towards people with ID in Cameroon and Ghana contexts, while attitudes towards people with intellectual disability among participants were found to be ambivalent. Additional findings were also reported in the manuscript. The authors provided elaborate background information and justification for this study, and I agree that covering the topic is important to better understand the situation of people with ID, especially in low and middle-income countries context, and to endorse the inclusion of the said people in society.

The followings are points that I would like to further discuss with the authors:

1. This study uses both CFA and PCA to explore the CLAS-ID underlying factors, however, both analyses yielded different results. The authors might want to discuss further the rationale of using both analyses in this study, and/or how the reader should interpret the results.

2. Page 11 [With respect to the Cameroonian data, the questionnaires were both in English and French since both languages are the medium of instruction and officially recognized languages]

Does this mean the Cameroonian participants receive two versions of the CLAS-ID, English and French? If yes, do the two versions are considered identical instruments? Please clarify.

3. Under the 'Predictors of Attitudes' subheading, there is a discussion about the association between the CLAS-ID sub-scales, which does not seems relevant with the sub-headings topic. Please clarify.

[page 19] 'the association between the sub-scales….

4. [page 2] From the perspectives of parents and individuals with IDs, empirical evidence suggests that they continue to suffer discrimination and rejection by other people in sub-Saharan African societies.

The authors might want to elaborate more on the examples of discrimination and rejection of people with ID in sub-Saharan societies. I think the information could help the readers to better understand the situation of people with ID in Ghana and Cameroon context.

5. [page 10] 'almost all the academics responded that the content was appropriate for data collection

Does this mean some academics disagree that the content of the instrument appropriate for data collection? If yes, then the author might want to provide more discussion about this.

6. The Author might want to add a reference for this statement:

[page 2] The onset of an ID is usually diagnosed before 18 years of age, with leading causes attributed to genetic factors and neurodevelopmental, neurological, and other medical conditions.

The finding regarding participants' ambivalent attitude towards people with ID, is not included in the abstract. The authors might want to consider adding the information in the abstract.

6. PLOS authors have the option to publish the peer review history of their article (what does this mean?). If published, this will include your full peer review and any attached files.

Reviewer #1: No

Reviewer #2: No

---

## [Decision Letter · Decision Letter 1]

3 Sep 2021

Cross-national Study of Communal Attitudes toward Individuals with Intellectual Disabilities in sub-Saharan Africa: Cameroon vs. Ghana

PONE-D-21-15410R1

Dear Dr. Opoku,

We’re pleased to inform you that your manuscript has been judged scientifically suitable for publication and will be formally accepted for publication once it meets all outstanding technical requirements.

Kind regards,

Robert Didden

Academic Editor

PLOS ONE

Additional Editor Comments (optional):

Reviewers' comments:

Reviewer's Responses to Questions

**Comments to the Author**

1. If the authors have adequately addressed your comments raised in a previous round of review and you feel that this manuscript is now acceptable for publication, you may indicate that here to bypass the “Comments to the Author” section, enter your conflict of interest statement in the “Confidential to Editor” section, and submit your "Accept" recommendation.

Reviewer #1: All comments have been addressed

Reviewer #2: All comments have been addressed

2. Is the manuscript technically sound, and do the data support the conclusions?

Reviewer #1: Yes

Reviewer #2: Yes

3. Has the statistical analysis been performed appropriately and rigorously? 

Reviewer #1: Yes

Reviewer #2: I Don't Know

4. Have the authors made all data underlying the findings in their manuscript fully available?

Reviewer #1: Yes

Reviewer #2: No

5. Is the manuscript presented in an intelligible fashion and written in standard English?

Reviewer #1: Yes

Reviewer #2: Yes

6. Review Comments to the Author

Reviewer #1: (No Response)

Reviewer #2: (No Response)

7. PLOS authors have the option to publish the peer review history of their article (what does this mean?). If published, this will include your full peer review and any attached files.

Reviewer #1: No

Reviewer #2: No

---

## [Editor Report · Acceptance letter]

17 Sep 2021

PONE-D-21-15410R1 

Cross-national Study of Communal Attitudes toward Individuals with Intellectual Disabilities in sub-Saharan Africa: Cameroon vs. Ghana

Dear Dr. Opoku:

I'm pleased to inform you that your manuscript has been deemed suitable for publication in PLOS ONE. Congratulations! Your manuscript is now with our production department. 

Kind regards, 

on behalf of

Professor Robert Didden 

Academic Editor

PLOS ONE